# Cryptanalysis and Improvement of a Chaotic Map-Based Image Encryption System Using Both Plaintext Related Permutation and Diffusion

**DOI:** 10.3390/e22050589

**Published:** 2020-05-24

**Authors:** Cheng-Yi Lin, Ja-Ling Wu

**Affiliations:** Department of Computer Science and Information Engineering, Graduate Institute of Networking and Multimedia, National Taiwan University, Taipei 106, Taiwan; sincerity@cmlab.csie.ntu.edu.tw

**Keywords:** image encryption, chaotic map, permutation, diffusion, cryptanalysis

## Abstract

In theory, high key and high plaintext sensitivities are a must for a cryptosystem to resist the chosen/known plaintext and the differential attacks. High plaintext sensitivity can be achieved by ensuring that each encrypted result is plaintext-dependent. In this work, we make detailed cryptanalysis on a published chaotic map-based image encryption system, where the encryption process is plaintext Image dependent. We show that some designing flaws make the published cryptosystem vulnerable to chosen-plaintext attack, and we then proposed an enhanced algorithm to overcome those flaws.

## 1. Introduction

With the rapid progress in digital technology and mobile devices, people have produced more and more user-generated information in these few years; besides texts, most of them are multimedia data, such as images and videos. With the increasing of information security and privacy protection issues, researchers have proposed lots of encryption algorithms [1] against unauthorized access to those user-generated media data. 

As suggested in [2], the main techniques used to develop image encryption algorithms can roughly be divided into the following six categories: chaotic map, DNA computing, cellular automata, wavelet transmission, neural networks, and compressive sensing. The extreme initial value sensitivity and high randomness of the chaotic systems make the chaotic maps the most popular tool in digital image encryption algorithms. This is because the chaotic systems have some useful properties, like being ergodic, highly sensitive to initial conditions, and pseudo-randomness, which fit the essential requirements for building a practical cryptosystem [3]. Fridrich [4] proposed the first image encryption algorithm based on a chaotic map in 1998. After that, a large number of digital image encryption algorithms that were based on chaotic maps were proposed [3], and the references therein. Since Liu et al. addressed lots of the chaotic map-based image encryption algorithms published on signal processing and information technology-related Journals before 2019, our following discussions will mainly focus on related works [5,6,7,8,9,10] posted on the Entropy Journal, most recently. 

In order to prevent an Image exchanging system from brute force and differential attacks, [5] presented a new image encryption mechanism, in which the Enhanced Logistic Map (ELM) and some simple encryption techniques, such as block scrambling, modified zigzag transformation, and chaotic-map based key generation, are used. The results of encryption are evaluated from six different security measures. The corresponding results demonstrate the security, reliability, efficiency, and flexibility of the proposed method. Sine-Tent map (STM) is intended in [7] to widen the chaotic range and to improve the shambolic performance of one-dimensional (1D) discrete chaotic maps. Based on STM, a novel double S-box based color image encryption algorithm is recommended, which offers better applicability in real-time image encryption. Notice that, in [7], the 256-bit hash value of a randomly sampled noise signal is applied to serve as the one-time initial values of the proposed system. Since there is only one operation, XOR, is used to diffuse the pixels; the encryption process can be executed very fast. [8] presents a chaotic-map based image encryption algorithm, where Logistic and Henon maps are used.

High key and high plaintext sensitivities are a must for a cryptosystem to resist the chosen/known plaintext and the differential attacks. High plaintext sensitivity can be achieved by ensuring that each encrypted result is plaintext-dependent. To reach this goal, [9] suggested that the surrounding of a plaintext image could be surrounded by a sequence generated from the SHA-256 hashed value of the corresponding plaintext. For conquering the same challenges, in [10], both the permutation and the diffusion stages of the proposed color image encryption scheme are related to the original plain image. For keeping high system efficiency, only one round of plaintext tied permutation and diffusion operations are performed for obtaining the cipher image. Moreover, the proposed approach can be applied to real-time image encryption directly since there is no need to send original image dependent security keys to the receiver. Our work is highly inspired by and related to [10]; we will explore it further in the next section.

In general, the analyses of encryption and decryption algorithms show that all of the algorithms, as mentioned above, have a good encryption effect, anti-attack ability, and high security. However, a minor designing flaw may make encryption algorithms vulnerable, even if they are chaotic-based. In the following, we will take the high plaintext sensitivity related work: “A simple Chaotic map-based Image Encryption System Using Both Plaintexts Related Permutation and Diffusion (CIES-UBPRPD)”, as proposed by Huang et al. [10], as an example to illustrate our above statement.

In a plain data-dependent cryptosystem, like CIES-UBPRPD, the cryptanalysis complexity is mostly increased. Therefore, the security level of the system will also be enhanced. Even though the experiments given in CIES-UBPRPD [10] showed lots of advantages when compared with conventional approaches, some designing flaws have been found by us. In this work, we first break a simplified version of CIES-UBPRPD with a chosen-plaintext attack, to demonstrate the effect of the discovered flaws. Subsequently, we make a few adjustments on CIES-UBPRPD and show that the modified version does relieve the defect of the original CIES-UBPRPD.

The rest of this paper is organized, as follows. Section 2 briefly describes the process of CIES-UBPRPD. In Section 3, we pointed out some design flaws of CIES-UBPRPD and demonstrated the effects of the weaknesses by issuing a chosen-plaintext attack against a simplified version of it. Afterwards, a modified version of CIES-UBPRPD is provided in Section 4. With the added supplements, the security level and the completeness of CIES-UBPRPD can be enlarged significantly. Some experimental results of the modified CIES-UBPRPD are presented in Section 5, in order to verify our previous claim, where the associated security analysis is also included. Finally, Section 6 concludes this writeup.

## 2. Related Work

In the original CIES-UBPRPD [10], all the arrays are started with index one, while in this work, we use an equivalent description but change all array indexes starting from zero. For the ease of discussion, except for the array indexes, most of our notations follow the usage that was adopted in [10].

### 2.1. The Involved Chaotic Maps

#### 2.1.1. Generalized Arnold’s Cat Map

Arnold’s Cat Map (ACM) is a well-known two-dimensional chaotic system proposed by the Russian mathematician Vladimir I. Arnold [11]. ACM is usually replaced by its generalized form to achieve higher security and higher randomness, as shown in Equation (1):(1)$$\left[\begin{array}{c} x^{\prime }\\ y^{\prime } \end{array}\right]=\left[\begin{array}{cc} 1 & a\\ b & ab+1 \end{array}\right]\left[\begin{array}{c} x\\ y \end{array}\right]\mathrm{mod}\left[\begin{array}{c} M\\ N \end{array}\right],$$
where (x,y) and $\left(x^{\prime },y^{\prime }\right)$ denote the positions of the original pixel and the target pixel, a and b are the system parameters, while M and N are the image’s height and width, respectively. After obtaining the target position $\left(x^{\prime },y^{\prime }\right)$, from Equation (1), the two pixels that are located at (x,y) and $\left(x^{\prime },y^{\prime }\right)$ will swap their pixel values with each other.

#### 2.1.2. Chebyshev Map

Chebyshev map [12] is a one-dimensional chaotic system that can be formulated, as shown in Equation (2):(2)$$x_{n+1}=T_{a}\left(x_{n}\right)=\cos\left(a\times \arccos x_{n}\right),$$
where $x_{n}\in \left[-1,1\right]$ and a∈ℕ is again one of the system parameters. For a≥2, chaotic behavior of Equation (2) holds. The initial value $x_{0}$ of the above equation is considered as part of the secret key. In CIES-UBPRPD, a is fixed at 4.

### 2.2. Image Encryption Algorithm

On the bases of ACM and Chebyshev map, the above-mentioned Chaotic map-based Image Encryption System—CIES-UBPRPD—was proposed by Huang et al. [10], where the most eye-catching property of the algorithm is its plaintext data-dependent encryption process. The encryption process of the original CIES-UBPRPD consists of the following two stages: 

#### 2.2.1. Permutation Stage

Step 1. Iterate the Chebyshev map defined in Equation (2) $M\times N+n_{0}+9\;$ times, discard the first $n_{0}$ terms to avoid the harmful effect, and obtain the chaotic sequence $x_{n}$, which contains (M×N+9) elements.
$$x_{n}=\left\{x_{0},x_{1},\cdots ,x_{M\times N+8}\right\}.$$

Step 2. Generate another sequence $x_{nq}$ according to
(3)$$x_{nq}\left(i\right)=\left(k_{1}⊗k_{2}⊗k_{3}\right)⊗x_{n}\left(i\right)\times 10^{15},$$
where i∈{0,1,⋯,8} and ⊗ denotes bitwise XOR operator.

Step 3. Calculate $sum_{r}$, $sum_{g}$, and $sum_{b}$ based on the following equations:(4)$$\begin{array}{c} sum_{r}=\overset{M-1}{\underset{i=0}{\sum }}\overset{N-1}{\underset{j=0}{\sum }}P_{R}\left(i,j\right),\\ sum_{g}=\overset{M-1}{\underset{i=0}{\sum }}\overset{N-1}{\underset{j=0}{\sum }}P_{G}\left(i,j\right),\\ sum_{b}=\overset{M-1}{\underset{i=0}{\sum }}\overset{N-1}{\underset{j=0}{\sum }}P_{B}\left(i,j\right), \end{array}$$
where $P_{R}$, $P_{G}$, and $P_{B}$ represent the Red, Green, and Blue channels of the plain image P, respectively.

Step 4. Calculate the system parameters by using the following equations:(5)$$\begin{array}{c} \{\begin{array}{l} b_{r}=\mathrm{mod}\left(x_{nq}\left(0\right)⊗sum_{r}+x_{nq}\left(1\right)⊗sum_{g}+x_{nq}\left(2\right)⊗sum_{b},\;256\right)\\ a_{r}=\mathrm{mod}\left(\left(b_{r}+1\right)\times \left(k_{1}⊗k_{2}⊗k_{3}\right),\;65536\right)+1 \end{array}\\ \{\begin{array}{l} b_{g}=\mathrm{mod}\left(x_{nq}\left(3\right)⊗sum_{r}+x_{nq}\left(4\right)⊗sum_{g}+x_{nq}\left(5\right)⊗sum_{b},\;256\right)\\ a_{g}=\mathrm{mod}\left(\left(b_{g}+1\right)\times \left(k_{1}⊗k_{2}⊗k_{3}\right),\;65536\right)+1 \end{array}\\ \{\begin{array}{l} b_{b}=\mathrm{mod}\left(x_{nq}\left(6\right)⊗sum_{r}+x_{nq}\left(7\right)⊗sum_{g}+x_{nq}\left(8\right)⊗sum_{b},\;256\right)\\ a_{b}=\mathrm{mod}\left(\left(b_{b}+1\right)\times \left(k_{1}⊗k_{2}⊗k_{3}\right),\;65536\right)+1, \end{array} \end{array}$$
where $\left(b_{r},a_{r}\right)$, $\left(b_{g},a_{g}\right)$, and $\left(b_{b},a_{b}\right)$ are pairs of parameters used to permute $P_{R}$, $P_{G}$, and $P_{B},$ respectively. Moreover, mod (*x,* m) denotes the calculation of “*x* mod m”.

Step 5. Permute $P_{R}$, $P_{G}$, and $P_{B}$ using the following modified Cat Map with the corresponding parameters:(6)$$\left[\begin{array}{c} x^{\prime }\\ y^{\prime } \end{array}\right]=\left[\begin{array}{cc} 1 & a\\ b+1 & a\left(b+1\right)+1 \end{array}\right]\left[\begin{array}{c} x+1\\ y+1 \end{array}\right]\mathrm{mod}\left[\begin{array}{c} M\\ N \end{array}\right],$$
where x∈{0,1,⋯,M−1} and y∈{0,1,⋯,N−1}. The scanning sequence is started from left to right and from top to bottom. After $P_{R}$, $P_{G}$ and $P_{B}$ are shuffled, we get the permuted image $P^{*}$.

#### 2.2.2. Diffusion Stage

Step 1. Transform $P_{R}^{*}$, $P_{G}^{*}$, and $P_{B}^{*},$ into three on-dimensional (1D) arrays $P_{R\_P}^{*}$, $P_{G\_P}^{*}$, and $P_{B\_P}^{*}$, respectively, by row-major ordering.

Step 2. Calculate the diffusion matrix D according to
(7)$$D\left(i\right)=\mathrm{mod}\left(\lfloor x_{n}\left(i+9\right)\times \left(k_{1}⊗k_{2}⊗k_{3})\right\rfloor ,\;256\right),$$
where i∈{0,1,⋯,M×N−1}.

Step 3. Calculate $C_{R\_P}$, $C_{G\_P}$, and $C_{B\_P}$ by using the following equations:(8)$$\{\begin{array}{l} C_{R\_P}\left(0\right)=\left(b_{r}+k_{1}\right)\mathrm{mod}256\\ C_{G\_P}\left(0\right)=\left(b_{g}+k_{2}\right)\mathrm{mod}256\\ C_{B\_P}\left(0\right)=\left(b_{b}+k_{3}\right)\mathrm{mod}256 \end{array}$$
(9)$$\begin{array}{c} \{\begin{array}{l} C_{R\_P}\left(i\right)=\mathrm{mod}\left(P_{R_{P}}^{*}\left(i\right)⊗D\left(i\right)+num,\;256\right)⊗C_{R}\left(i-1\right)\;\\ C_{G\_P}\left(i\right)=\mathrm{mod}\left(P_{G_{P}}^{*}\left(i\right)⊗D\left(i\right)+num,\;256\right)⊗C_{G}\left(i-1\right)\\ C_{B\_P}\left(i\right)=\mathrm{mod}\left(P_{B_{P}}^{*}\left(i\right)⊗D\left(i\right)+num,\;256\right)⊗C_{B}\left(i-1\right) \end{array} \end{array}$$
where $num=\left(a_{r}\times b_{r}+a_{g}\times b_{g}+a_{b}\times b_{b}\right)⊗\left(k_{1}+k_{2}+k_{3}\right)$, and i∈{1,2,⋯,M×N−1}.

Step 4. Transform $C_{R\_P}$, $C_{G\_P},$ and $C_{B\_P}$ into three grayscale images with size M×N, and then merge them into one color cipher image C, with size M×N×3.

Notice that the permutation processes executed in Step 5 involved parameters $sum_{r}$, $sum_{g,}$ and $sum_{b}$, which are input (plaint) image data-dependent (cf. Equations (4) and (5)). Notice that, as pre-described in Section 1, the encryption process only includes one round permutation stage and diffusion stage. Conceptually and theoretically, if the encryption process of a cryptosystem is plaintext data-dependent, the associated cryptanalysis is much complicated. Therefore, the corresponding security level of the system is much enhanced, as compared with its data-independent counterpart.

### 2.3. Image Decryption Algorithm

Similarly, also from [10], the decryption process of the original CIES-UBPRPD consists of the following five steps:

Step 1. Transform $C_{R}$, $C_{G},$ and $C_{G}$ into three 1D arrays $C_{R\_P}$, $C_{G\_P},$ and $C_{B\_P}$, respectively, by row-major ordering.

Step 2. Calculate the diffusion matrix $D=\left\{d_{0},d_{1},\cdots ,d_{M\times N-1}\right\}$ based on Equation (7).

Step 3. Calculate the system parameters by using the following equations:(10)$$\{\begin{array}{l} b_{r}=\left(C_{R\_P}\left(0\right)-k_{1}\right)\mathrm{mod}256\\ b_{g}=\left(C_{G\_P}\left(0\right)-k_{2}\right)\mathrm{mod}256\\ b_{b}=\left(C_{B\_P}\left(0\right)-k_{3}\right)\mathrm{mod}256 \end{array}$$
(11)$$\begin{array}{l} a_{r}=\mathrm{mod}\left(\left(b_{r}+1\right)\times \left(k_{1}⊗k_{2}⊗k_{3}\right),\;65536\right)+1\\ a_{g}=\mathrm{mod}\left(\left(b_{g}+1\right)\times \left(k_{1}⊗k_{2}⊗k_{3}\right),\;65536\right)+1\;\\ a_{b}=\mathrm{mod}\left(\left(b_{b}+1\right)\times \left(k_{1}⊗k_{2}⊗k_{3}\right),\;65536\right)+1. \end{array}$$

Step 4. Reconstruct $P_{R\_P}^{*}$, $P_{G\_P}^{*},$ and $P_{B\_P}^{*}$ according to
(12)$$\{\begin{array}{l} P_{R_{P}}^{*}\left(i\right)=\mathrm{mod}\left(\left(C_{R_{P}}\left(i\right)⊗C_{R_{P}}\left(i-1\right)\right)-num,\;256\right)⊗D\left(i\right)\\ P_{G_{P}}^{*}\left(i\right)=\mathrm{mod}\left(\left(C_{G_{P}}\left(i\right)⊗C_{G_{P}}\left(i-1\right)\right)-num,\;256\right)⊗D\left(i\right)\\ P_{B_{P}}^{*}\left(i\right)=\mathrm{mod}\left(\left(C_{B_{P}}\left(i\right)⊗C_{B_{P}}\left(i-1\right)\right)-num,\;256\right)⊗D\left(i\right), \end{array}$$
where
(13)$$num=\left(a_{r}\times b_{r}+a_{g}\times b_{g}+a_{b}⊗b_{b}\right)⊗\left(k_{1}+k_{2}+k_{3}\right),$$
and i∈{1,2,⋯,M×N−1}. Then transform these three arrays into 2D arrays $P_{R}^{*}$, $P_{G}^{*},$ and $P_{B}^{*}$, respectively.

Step 5. Reconstruct $P_{R}$, $P_{G},$ and $P_{B}$ by using the Cat Maps defined in Equation (1), but now the scanning sequence is started from right to left and from bottom to top.

## 3. Cryptanalysis

Cryptanalysis is a must procedure for any cryptosystem to be applied to any real applications. We launched a few analyses on CIES-UBPRPD when we learned it from [10]. This section is organized, as follows. Section 3.1 reports the security weaknesses we found and Section 3.2 presents the attack that we used to break a simplified version of the original CIES-UBPRPD.

### 3.1. Security Weaknesses

#### 3.1.1. Equivalent Classes in Keyspace

After detailed analyses of CIES-UBPRPD, we found that two secret key groups $key_{1}=\left(x_{0},k_{1},k_{2},k_{3},n_{0}\right)$ and $key_{2}=\left(x_{0}^{\prime },k_{1}^{\prime },k_{2}^{\prime },k_{3}^{\prime },n_{0}^{\prime }\right)$ could play indistinguishable roles to each other, in the original cryptosystem, if the following conditions are satisfied:(14)$$\{\begin{array}{l} n_{0}=n_{0}^{\prime }\\ x_{0}=x_{0}^{\prime }\\ k_{1}⊗k_{2}⊗k_{3}=k_{1}^{\prime }⊗k_{2}^{\prime }⊗k_{3}^{\prime }\;\\ k_{1}\equiv k_{1}^{\prime }\left(\mathrm{mod}256\right)\\ k_{2}\equiv k_{2}^{\prime }\left(\mathrm{mod}256\right)\\ k_{3}\equiv k_{3}^{\prime }\left(\mathrm{mod}256\right). \end{array}$$

For demonstration, here we choose $key_{1}=\left(0.7,784533,763092,777777,1500\right)$ and $key_{2}=\left(0.7,353173,676820,307761,1500\right)$, which satisfy all of the conditions given in Equation (14). First, we use $key_{1}$ to encrypt the benchmark image Lena (left, Figure 1) and obtain the corresponding cipher image (middle, Figure 1). Subsequently, we use $key_{2}$ to decrypt the cipher image and obtain the recovered image (right, Figure 1). 

The above example implies that one can split the set of all keys into equivalence classes that are based on the conditions given in Equation (14), and the effects of keys belonging to the same equivalence classes will be indistinguishable to each other in CIES-UBPRPD. This property shrinks the effective keyspace from $(10^{16}\times \left(10^{12}-10^{5}{)}^{3}\times 1500\right)\approx 2^{183}$ to $\left(10^{16}\times 2^{8}\times 2^{8}\times 2^{8}\times 2^{32}\times 1500\right)\approx 2^{120}$ (since $10^{12}\approx 2^{40}$, we can assume $k_{1}$, $k_{2}$ and $k_{3}$ are of 40-bit long), which is far less than what [10] initially claimed.

#### 3.1.2. Low Sensitivity to the Change of Plaintext

Once the secret key group is chosen, and the summation of pixel values in each channel is given, then the parameters used in the Cat Map are always fixed. Having the same settings means that the permutation mapping will be fixed no matter what the plain image is. Furthermore, there is no cross-channel interaction during permutation and diffusion stages in the original CIES-UBPRPD, which suggests that errors inside one channel will not propagate to the other channels.

Therefore, we can construct two similar plain images where only their R channels are different, but their sum of R channel remains the same. Additionally the corresponding cipher images of these two images will have no differences in G and B channels. This situation violates the diffusion property that a chaotic-map based cryptosystem is looking for.

For demonstration, we modify the standard Lena image by increasing the first-pixel value by 1 and decreasing the second-pixel value also by 1 in the R channel, and then compare the corresponding cipher image with that of the standard Lena image. The results are shown in Figure 2 and Table 1, the details of two widely used measures, number of pixels change rate (NPCR) and unified average changing intensity (UACI), will be given in Section 5.2.5.

### 3.2. Chosen-Plaintext Attack

As stated by Bruce Schneier [13], in academic cryptography, a weakness or a break in a scheme is usually defined quite conservatively: it might require impractical amounts of time, memory, or known plaintexts. It also might require the attacker to be able to do things many real-world attackers cannot. For example, the attacker might need to choose particular plaintexts to be encrypted or even to ask for plaintexts to be encrypted while using several keys related to the secret key. Furthermore, it might only reveal a small amount of information, enough to prove the cryptosystem imperfect, but too little to be useful to real-world attackers. Finally, an attack might only apply to a weakened version of cryptographic tools, like a reduced-round block cipher, as a step towards breaking the full system.

Following the principles of cryptanalysis stated above, we now present a chosen-plaintext attack that works if the size of all images equals 256×256 pixels. 

#### 3.2.1. Extraction of the Permutation Matrix

Construct a special plain image P, such that
$$P_{R}=P_{G}=P_{B}={\left[\begin{array}{cccc} 1 & 1 & \cdots & 1\\ 1 & 1 & \cdots & 1\\ \vdots & \vdots & \ddots & \vdots \\ 1 & 1 & \cdots & 1 \end{array}\right]}_{256\times 256}.$$Encrypt P using CIES-UBPRPD to obtain the corresponding cipher image C. We denote the image after permutation stage as $P^{*}$ (an intermediate product during the execution of the whole encryption process).Let us use R channel as an example: select two different positions (a,b) and (x,y), where a,b,x,y∈{0,1,⋯,255} and construct another special plain image $P^{\prime }$, such that
$$\begin{array}{c} {P^{\prime }}_{R}={\left[f\left(i,j\right)\right]}_{256\times 256},\\ {P^{\prime }}_{G}={P^{\prime }}_{B}={\left[\begin{array}{cccc} 1 & 1 & \cdots & 1\\ 1 & 1 & \cdots & 1\\ \vdots & \vdots & \ddots & \vdots \\ 1 & 1 & \cdots & 1 \end{array}\right]}_{256\times 256} \end{array},$$
where
$$f\left(i,j\right)=\{\begin{array}{l} 0,\;if\;\left(i,j\right)=\left(a,b\right)\\ 2,\;if\;\left(i,j\right)=\left(x,y\right)\\ 1,\;else \end{array}.$$Encrypt $P^{\prime }$ to obtain its cipher image $C^{\prime }$. From Section 3.1.2, we knew that P and $P^{\prime }$ share the same parameters used in ACM; thus, they have the same permutation mapping. Let us denote the first position of different values that occurred in $C_{R}$ and ${C^{\prime }}_{R}$, following the raster-scan order, as ΔC. It means that ${P^{\prime *}}_{R}\left(\Delta C\right)\neq P_{R}^{*}\left(\Delta C\right)=1$, such that ${C^{\prime }}_{R}\left(\Delta C\right)\neq C_{R}\left(\Delta C\right)$, which means either ${P^{\prime }}_{R}\left(a,b\right)=0$ or ${P^{\prime }}_{R}\left(x,y\right)=2$ will be moved to the position ΔC after performing the permutation stage. This property reveals some information regarding the permutation behavior of CIES-UBPRPD. We define ((a,b),(x,y),ΔC) as a tuple of constraints.Choose a different (a,b) or (x,y), repeat Step 3 to Step 4 several times and collect all of the produced constraint tuples, and then define the associated collection of constraint tuples as a set S.Construct a 256×256 matrix Z, by setting the chosen positions (*a*, *b*) and (*x*, *y*), used in Step 2, with different positive integers and all other positions with value 0. For example, assume there are two constraint tuples ((0,0),(0,1),(8,8)) and ((0,0),(0,2),(6,9)) in S, we can now set Z(0,0)=1, Z(0,1)=2, Z(0,2)=3, and Z(i,j)=0, ∀(i,j)∉{(0,0),(0,1),(0,2)}.Using the brute-force searching algorithm, denoted as Algorithm 1 in the following, to find $b_{r}$ and $\hat{a_{r}}$ for all of the above chosen plain images, where $\hat{a_{r}}=a_{r}\mathrm{mod}256$. In the specifically considered case, all of the images are of size 256×256×3, $a_{r},$ and $\hat{a_{r}}$ are equivalent in the permutation stage. Actually, we do not need to know what $a_{r}$ exactly is, but only its last eight bits. If Algorithm 1 outputs more than one candidate pair, let us go back to Step 3 to collect more constrained tuples and iterate this step until only one pair is left.Making changes to G and B channels, repeat the procedures listed in Step 3 to Step 7 and obtain $b_{g}$, $\hat{a_{g}}$, $b_{b}$, and $\hat{a_{b}}$, where $\hat{a_{g}}=a_{g}\mathrm{mod}256$ and $\hat{a_{b}}=a_{b}\mathrm{mod}256$.Extract $k_{1}\mathrm{mod}256$, $k_{2}\mathrm{mod}256$, and $k_{3}\mathrm{mod}256$ by using
$$\{\begin{array}{l} k_{1}\mathrm{mod}256=\left(C_{R}\left(0,0\right)-b_{r}\right)\mathrm{mod}256\\ k_{2}\mathrm{mod}256=\left(C_{G}\left(0,0\right)-b_{g}\right)\mathrm{mod}256\\ k_{3}\mathrm{mod}256=\left(C_{B}\left(0,0\right)-b_{b}\right)\mathrm{mod}256 \end{array}.$$We denote these three values as $\hat{k_{1}}$, $\hat{\;k_{2}},$ and $\hat{k_{3}}$, respectively, for convenience.

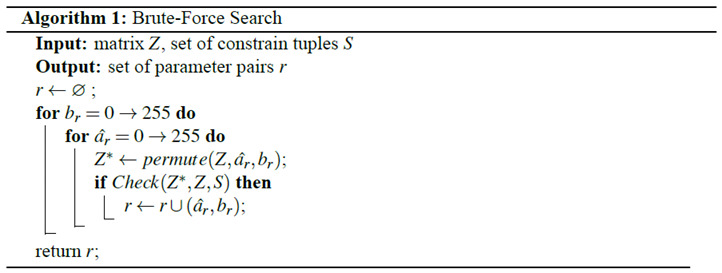


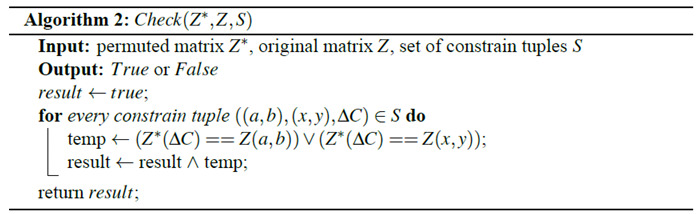



#### 3.2.2. Extraction of the Diffusion Matrix

First, we convert above-mentioned $C_{R}$ into 1D array $C_{R\_P}$ by row major ordering. We can now obtain the diffusion matrix D, according to
$$\begin{array}{rr} D\left(i\right) & =\mathrm{mod}\left(\left(C_{R\_P}\left(i\right)⊗C_{R\_P}\left(i-1\right)\right)-{num}^{\prime },256\right)⊗P_{R\_P}^{*}\left(i\right)\\ & =\mathrm{mod}\left(\left(C_{R_{P}}\left(i\right)⊗C_{R_{P}}\left(i-1\right)\right)-{num}^{\prime },256\right)⊗1,\; \end{array}$$
where
$${num}^{\prime }=\left(\hat{a_{r}}\times b_{r}+\hat{a_{g}}\times b_{g}+\hat{a_{b}}\times b_{b}\right)⊗\left(\hat{k_{1}}+\hat{k_{2}}+\hat{k_{3}}\right),$$

i∈{1,2,⋯,M×N−1}, and $P_{R\_P}^{*}$ is a 1D array transformed from $P_{R}^{*}$ by row major ordering. Notice that the first element of the diffusion matrix is not used in the cryptosystem, so we can just assign D(0)=0.

So far, we have already extracted $\hat{k_{1}}$, $\hat{k_{2}}$, $\hat{k_{3},}$ and the diffusion matrix D, which are all the required information for decrypting the cipher image.

#### 3.2.3. Recovering the Original Plain Image

Assume that the cipher image is $\bar{C}$ with the size 256×256×3, and the attacker already extracts $\hat{k_{1}}$, $\hat{k_{2}}$, $\hat{k_{3}}$, and the diffusion matrix D according to the analyses given above. The attacker can decrypt $\bar{C}$ using the following steps:

Transform $\bar{C_{R}}$, $\bar{C_{G}}$, and $\bar{C_{B}}$ into three 1D arrays $\bar{C_{R\_P}}$, $\bar{C_{G\_P}}$, and $\bar{C_{B\_P}}$, respectively, in row major ordering.Calculate the required parameters, as follows.
$$\begin{array}{c} \begin{array}{c} \{\begin{array}{l} b_{r}=\left(\bar{C_{R\_P}}\left(0\right)-\hat{k_{1}}\right)\mathrm{mod}256\\ b_{g}=\left(\bar{C_{G\_P}}\left(0\right)-\hat{k_{2}}\right)\mathrm{mod}256\\ b_{b}=\left(\bar{C_{B\_P}}\left(0\right)-\hat{k_{3}}\right)\mathrm{mod}256 \end{array} \end{array}\\ \{\begin{array}{l} \hat{a_{r}}=\mathrm{mod}\left(\left(b_{r}+1\right)\times \left(\hat{k_{1}}⊗\hat{k_{2}}⊗\hat{k_{3}}\right),256\right)+1\\ \hat{a_{g}}=\mathrm{mod}\left(\left(b_{g}+1\right)\times \left(\hat{k_{1}}⊗\hat{k_{2}}⊗\hat{k_{3}}\right),256\right)+1\\ \hat{a_{b}}=\mathrm{mod}\left(\left(b_{b}+1\right)\times \left(\hat{k_{1}}⊗\hat{k_{2}}⊗\hat{k_{3}}\right),256\right)+1. \end{array} \end{array}$$Reconstruct $\bar{P_{R\_P}^{*}}$, $\bar{P_{G\_P}^{*}}$ and $\bar{P_{B\_P}^{*}}$ according to
$$\{\begin{array}{l} \bar{P_{R\_P}^{*}}=\mathrm{mod}\left(\left(\bar{C_{R\_P}}\left(i\right)⊗\bar{C_{R\_P}}\left(i-1\right)\right)-{num}^{\prime },256\right)⊗D\left(i\right)\\ \bar{P_{G\_P}^{*}}=\mathrm{mod}\left(\left(\bar{C_{G\_P}}\left(i\right)⊗\bar{C_{G\_P}}\left(i-1\right)\right)-{num}^{\prime },256\right)⊗D\left(i\right)\\ \bar{P_{B\_P}^{*}}=\mathrm{mod}\left(\left(\bar{C_{B\_P}}\left(i\right)⊗\bar{C_{B\_P}}\left(i-1\right)\right)-num^{\prime },256\right)⊗D\left(i\right), \end{array}$$
where
$${num}^{\prime }=\left(\hat{a_{r}}\times b_{r}+\hat{a_{g}}\times b_{g}+\hat{a_{b}}\times b_{b}\right)⊗\left(\hat{k_{1}}+\hat{k_{2}}+\hat{k_{3}}\right),$$
and i∈{1,2,⋯,M×N−1}. Subsequently, transform these three arrays into two-dimensional (2D) arrays $\bar{P_{R}^{*}}$, $\bar{P_{G}^{*}}$, and $\bar{P_{B}^{*}}$, respectively.Reconstruct $\bar{P_{R}}$, $\bar{P_{G}},$ and $\bar{P_{B}}$ by using ACM, but now the scanning sequence is from right to left and from bottom to top.

We rescale the standard Lena to 256×256×3 and encrypt it with CIES-UBPRPD, and then crack the cipher image with the proposed chosen-plaintext attack. Figure 3 shows the simulation result.

## 4. Improved CIES-UBPRPD Algorithm

### 4.1. The Weaknesses of the Original CIES-UBPRPD

We can crack the original CIES-UBPRPD by the chosen-plaintext attack comes from its following weaknesses:

Misuse of the modulo operation. A value’s remainder divided by 256 equals to the last eight bits in its binary representation. This operation makes the last eight bits of the value more important than the rest parts. This unequal importance in bits gives us a large number of clues for finding the equivalent classes of parameters in CIES-UBPRPD.The parameters used in ACM are not very sensitive to the initial plain images. As we pointed out in Section 3.1.2, images that have the same $sum_{r}$, $sum_{g}$, and $sum_{b}$ share the same system parameters. Thus, it is vulnerable to differential attacks.The diffusion matrix (process) depends only on secret keys, but not the plain images. Once we cracked one cipher image and extracted the diffusion matrix, we can use it to decrypt the other cipher images.

### 4.2. The Enhanced CIES-UBPRPD

In the enhanced encryption algorithm, instead of the summations of pixel values in each channel, the corresponding SHA-256 hashed values are used as one of the features of a plain image, and this replacement still makes the associated diffusion matrix plaintext-dependent. SHA-256 is a secure cryptographic hash that belongs to SHA-2 families. The hash value served as an external secret key; however, it is dangerous to reuse the same external key when encrypting the same image. Therefore, we add a random number with the precision of $10^{-16}$ as an additional input to SHA-256 each time that we calculate the hash value. In this way, we can use the hashing output as a one-time key.

#### 4.2.1. Secret Key Formulation

There are six secret keys in the proposed enhanced CIES-UBPRPD algorithm, including the external secret key H that is generated from SHA-256, the initial value $x_{0}$ of Chebyshev map, and the four positive integers $k_{1}$, $k_{2}$, $k_{3}$, and $n_{0}$, where H is a 256-bit binary number, $x_{0}\in \left(0,1\right),\;k_{1}\in \left[10^{5}\ldots 10^{12}],k_{2}\in [10^{5}\ldots 10^{12}],k_{3}\in [10^{5}\ldots 10^{12}\right]$, and $n_{0\;}\in \left[1000,\;2500\right]$. H is then divided into 32 8-bit blocks as $H=h_{0},h_{1},\cdots ,h_{31}$.

#### 4.2.2. Image Encryption Algorithm

##### Permutation Stage

Use $x_{0}$ as the initial value and iterate the Chebyshev map $\left(n_{0}+131\right)$ times, discard the first $n_{0}$ elements to avoid the harmful effect, and obtain the chaotic sequences $x_{n}$, which contain 131 elements. That is,
$$x_{n}=\left\{x_{0},x_{1},\cdots ,x_{130}\right\}.$$Generate another sequence $x_{nq}$ by
$$x_{nq}\left(i\right)=\lfloor x_{i}\times k_{i\;mod\;3}\times \cos h_{i\;mod\;32}\rfloor \;mod\;256,\;\mathrm{where}\;i\in \left\{0,1,\cdots ,130\right\}.$$Calculate the parameters by following equations:$$\begin{array}{c} a=\left(\overset{31}{\underset{i=0}{\sum }}h_{i}\times x_{nq}\left(i\right)\right)\mathrm{mod}65536\\ b=\left(\overset{31}{\underset{i=0}{\sum }}h_{i}\times x_{nq}\left(i+32\right)\right)\mathrm{mod}65536\\ c=\left(\overset{31}{\underset{i=0}{\sum }}h_{i}\times x_{nq}\left(i+64\right)\right)\mathrm{mod}65536\\ d=\left(\overset{31}{\underset{i=0}{\sum }}h_{i}\times x_{nq}\left(i+96\right)\right)\mathrm{mod}65536 \end{array}$$Permute P using the following three-dimensional (3D) cat map:(15)$$\left[\begin{array}{c} i^{\prime }\\ j^{\prime }\\ k^{\prime } \end{array}\right]=\left[\begin{array}{ccc} 1 & a & 0\\ b & ab+1 & 0\\ c & d & 1 \end{array}\right]\left[\begin{array}{c} i\\ j\\ k \end{array}\right]\mathrm{mod}\left[\begin{array}{c} M\\ N\\ 3 \end{array}\right],$$
where i∈{0,1,⋯,M−1}, j∈{0,1,⋯,N−1}, and k∈{0,1,2} is used to indicate the color channel. The scanning sequence is from R channel to B channel, from left to right and from top to bottom. After this step, we get the permuted image $P^{*}$.

##### Diffusion Stage

Transform $P_{R}^{*}$, $P_{G}^{*}$, and $P_{B}^{*},$ into three 1D arrays $P_{R\_P}^{*}$, $P_{G\_P}^{*}$ and $P_{B\_P}^{*}$ by row-major ordering.Use $y_{0}=\frac{\cos a+\cos b+\cos c+\cos d+x_{0}}{5}$ as the new initial value and iterate the Chebyshev map $\left(3\times M\times N+n_{0}\right)$ times, discard the first n_0_ elements and generate another chaotic sequence $y_{n}$, which contains 3×M×N elements. That is,
$$y_{n}=\left\{x_{0},x_{1},\cdots ,x_{3\times M\times N-1}\right\}.$$Calculate the diffusion matrices $D_{R}$, $D_{G}$, and $D_{B}$ according to:$$\{\begin{array}{c} D_{R}\left(i\right)=\lfloor y_{i}\times \left(k_{2}⊗k_{3}\right)\rfloor \;mod\;256\\ D_{G}\left(i\right)=\lfloor y_{i+MN}\times \left(k_{1}⊗k_{3}\right)\rfloor \;mod\;256\\ D_{B}\left(i\right)=\lfloor y_{i+2MN}\times \left(k_{1}⊗k_{2}\right)\rfloor \;mod\;256 \end{array}\;\mathrm{where}\;i\in \left\{0,1,\cdots ,M\times N-1\right\}.$$Calculate $C_{R\_P}$, $C_{G\_P}$ and $C_{B\_P}$ by using
$$\begin{array}{c} \{\begin{array}{l} C_{R\_P}\left(0\right)=\mathrm{mod}\left(P_{R\_P}^{*}\left(0\right)⊗D_{R}\left(0\right)+num,\;256\right)⊗x_{nq}\left(128\right)\\ C_{G\_P}\left(0\right)=\mathrm{mod}\left(P_{G\_P}^{*}\left(0\right)⊗D_{G}\left(0\right)+num,\;256\right)⊗x_{nq}\left(129\right)\\ C_{B\_P}\left(0\right)=\mathrm{mod}\left(P_{B\_P}^{*}\left(0\right)⊗D_{B}\left(0\right)+num,\;256\right)⊗x_{nq}\left(130\right) \end{array}\\ \{\begin{array}{l} C_{R\_P}\left(i\right)=\mathrm{mod}\left(P_{R\_P}^{*}\left(i\right)⊗D_{R}\left(i\right)+num,\;256\right)⊗C_{B}\left(i-1\right)\\ C_{G\_P}\left(i\right)=\mathrm{mod}\left(P_{G\_P}^{*}\left(i\right)⊗D_{G}\left(i\right)+num,\;256\right)⊗C_{R}\left(i\right)\\ C_{B\_P}\left(i\right)=\mathrm{mod}\left(P_{B\_P}^{*}\left(i\right)⊗D_{B}\left(i\right)+num,\;256\right)⊗C_{G}\left(i\right), \end{array} \end{array}$$
where $num=\left(\left(a+b+c+d\right)⊗\left(k_{1}+k_{2}+k_{3}\right)\right)\mathrm{mod}256$ and i∈{1,2,⋯,M×N−1}.Transform $C_{R\_P}$, $C_{G\_P},$ and $C_{B\_P}$ into three grayscale images with size M×N, and then merge them into color cipher image C with size M×N×3.

##### Image Decryption Algorithm

Transform $C_{R}$, $C_{G},$ and $C_{G}$ into three 1D arrays $C_{R\_P}$, $C_{G\_P},$ and $C_{B\_P}$ respectively, by row-major ordering.Calculate the chaotic sequence $x_{nq}$ in the same way as the encryption process.Calculate the parameters a,b,c,d in the same way as the encryption process.Calculate the diffusion matrices $D_{R}$, $D_{G},$ and $D_{B}$ in the same way as the encryption process.Reconstruct $P_{R\_P}^{*}$, $P_{G\_P}^{*},$ and $P_{B\_P}^{*}$ by using
$$\begin{array}{c} \{\begin{array}{l} P_{R\_P}^{*}\left(0\right)=\mathrm{mod}\left(\left(C_{R\_P}\left(0\right)⊗x_{nq}\left(128\right)\right)-num,\;256\right)⊗D_{R}\left(0\right)\\ P_{G\_P}^{*}\left(0\right)=\mathrm{mod}\left(\left(C_{G\_P}\left(0\right)⊗x_{nq}\left(129\right)\right)-num,\;256\right)⊗D_{G}\left(0\right)\\ P_{B\_P}^{*}\left(0\right)=\mathrm{mod}\left(\left(C_{B\_P}\left(0\right)⊗x_{nq}\left(130\right)\right)-num,\;256\right)⊗D_{B}\left(0\right) \end{array}\\ \{\begin{array}{l} P_{R\_P}^{*}\left(i\right)=\mathrm{mod}\left(\left(C_{R\_P}\left(i\right)⊗C_{R\_P}\left(i-1\right)\right)-num,\;256\right)⊗D_{R}\left(i\right)\\ P_{G\_P}^{*}\left(i\right)=\mathrm{mod}\left(\left(C_{G\_P}\left(i\right)⊗C_{G\_P}\left(i-1\right)\right)-num,\;256\right)⊗D_{G}\left(i\right)\\ P_{B\_P}^{*}\left(i\right)=\mathrm{mod}\left(\left(C_{B\_P}\left(i\right)⊗C_{B\_P}\left(i-1\right)\right)-num,\;256\right)⊗D_{B}\left(i\right) \end{array} \end{array},$$
where $num=\left(\left(a+b+c+d\right)⊗\left(k_{1}+k_{2}+k_{3}\right)\right)\mathrm{mod}256$ and i∈{1,2,⋯,M×N−1}. Subsequently, transform these three arrays into 2D arrays $P_{R}^{*}$, $P_{G}^{*}$ and $P_{B}^{*},$ respectively.Reconstruct $P_{R}$, $P_{G},$ and $P_{B}$ by using 3D cat map in Equation (15), but now the scanning sequence is from B channel to R channel, from right to left, and from bottom to top.

## 5. Experimental Results

### 5.1. Verification of Encryption and Decryption Algorithms

We apply the proposed algorithm to several testing images (all of size 512×512×3) to demonstrate the algorithm’s performance. The secret keys are set, as follows: $x_{0}=0.3,k_{1}=111111,k_{2}=222222,k_{3}=333333,$ and $n_{0}=1000$. Figure 4 shows the encryption and decryption results. From the results, we can say that the cipher images are noise-like and irrelevant to the plain images.

### 5.2. Security Analyses

#### 5.2.1. KeySpace Analysis

Keyspace is defined as the cardinality of the set of all possible keys. Having a large keyspace is an important factor for ensuring a cryptosystem to resist the brute force attack. The best-known attack complexity of SHA-256 is in the order of $2^{128}$. The range of the rest keys are: $x_{0}\in \left(0,1\right),k_{1},k_{2},k_{3}\in \left[10^{5}...10^{12}\right]$ and $n_{0}\in \left[1000,\;2500\right]$. If $x_{0}$ has the precision of $10^{-16}$, the keyspace of the proposed scheme can reach to $2^{128}\times 10^{16}\times \left(10^{12}-10^{5}\right)\times \left(10^{12}-10^{5}\right)\times \left(10^{12}-10^{5}\right)\times 1500\approx 2^{311}$, which is much larger than $2^{100}$ and enough to make the brute force attack invalid.

#### 5.2.2. Histogram Analysis

An image’s histogram reflects the distribution of its pixels’ intensity values, which reveals some statistical information of the image to attackers. To against statistical attacks, the histogram of cipher images generated from a secure encryption system should be flat. The distributions of cipher images’ histograms are close to uniformly, indicating that the cipher images are nearly random, and it is rather tough to retrieve any useful statistical information from them, as we can see in Figure 5 and Figure 6.

#### 5.2.3. Correlation Analysis

In a natural image (or plain image), two adjacent pixels usually have strong correlation with each other. In contrast, the correlation coefficient of a cipher image should be decreased to zero in order to prevent statistical attacks. We use Equation (16) to measure the correlation of all adjacent pixels at horizontal, vertical, diagonal, and anti-diagonal directions:(16)$$r_{xy}=\frac{cov\left(x,y\right)}{\sqrt{D\left(x\right)}\sqrt{D\left(y\right)}},$$

Here,
$$\begin{array}{c} cov\left(x,y\right)=\frac{1}{N}\sum_{i=1}^{N}(x_{i}-E\left(x\right))\left(y_{i}-E\left(y\right)\right),\\ D\left(x\right)=\frac{1}{N}\sum_{i=1}^{N}{\left(x_{i}-E\left(x\right)\right)}^{2},\\ E\left(x\right)=\frac{1}{N}\sum_{i=1}^{N}x_{i}, \end{array}$$

x and y are the two adjacent pixel values and N is the number of pairs of adjacent pixels.

The correlation coefficients of all plain images are close to 1, while those of cipher images are nearly 0, as shown in Table 2. Furthermore, we randomly select 2000 pairs of adjacent pixels at the four specific directions from the R channel of the standard Lena image and its corresponding cipher image, and then plot the scatter diagrams in Figure 7.

#### 5.2.4. Key Sensitivity Analysis

A cryptosystem might suffer from differential attacks if cipher images that are generated from different keys are similar. Thus, a secure encryption algorithm should be highly sensitive to all keys, which means even a tiny change in the secret key will lead to a completely different cipher image.

Figure 4 illustrates the plain, cipher, and decrypted Lena images. We change $H,\;x_{0},k_{1},k_{2},k_{3}$, and $n_{0}$ by one bit (i.e., $10^{-16}$ for $x_{0}$ and 1 for $H,k_{1},k_{2},k_{3}$ and $n_{0}$) to obtain six new cipher images, and the results are shown in Figure 8. Figure 9 shows the differential results between these six cipher images and the cipher image obtained in Figure 4b. Furthermore, we use the six one-bit difference key groups to decrypt Figure 4b, and the decrypting results are shown in Figure 10.

#### 5.2.5. Plaintext Sensitivity Analysis

Similar to key sensitivity, a secure encryption algorithm should also be very sensitive to plain images. We use NPCR (number of pixels change rate) and UACI (unified average changing intensity) to measure the difference between two cipher images, which are defined, as follows:$$\begin{array}{c} NPCR=\frac{1}{M\times N}\sum_{i=0}^{M-1}\sum_{j=0}^{N-1}D\left(i,j\right)\times 100\%,\\ UACI=\frac{1}{M\times N}\sum_{i=0}^{M-1}\sum_{j=0}^{N-1}\frac{\left|c_{1}\left(i,j\right)-c_{2}\left(i,j\right)\right|}{255}\times 100\%, \end{array}$$
where
$$D\left(i,j\right)=\{\begin{array}{l} 0,\;if\;c_{1}\left(i,j\right)=c_{2}\left(i,j\right)\\ 1,\;if\;c_{1}\left(i,j\right)\neq c_{2}\left(i,j\right) \end{array},$$
and $c_{1}$ and $c_{2}$ are two cipher images. The theoretical values of NPCR and UACI between two different cipher images are 99.6094% and 33.4635%, respectively.

We evaluate plaintext sensitivity, as follows. First, we randomly select $x_{0},k_{1},k_{2},k_{3},n_{0}$ from the keyspace, encrypt the test image $p_{1},$ and with it to get the cipher image $c_{1}$. Second, we randomly select a position in $p_{1}$, increasing each channel’s value by 1 in some places to obtain a modified image $p_{2}$. Next, we encrypt $p_{2}$ to obtain a cipher image $c_{2}$. Subsequently, calculate the NPCR and UACI values between $c_{1}$ and $c_{2}$. Repeat this process 200 times and list the averaged NPCR and UACI values in Table 3. Both the averaged NPCR and UACI values for all tested images are very close to their theoretical optimal ones, which means the enhanced CIES-UBPRPD algorithm do sensitive to the plain input images, as shown in Table 3.

### 5.3. Robustness Analyses

Cipher images can easily be polluted during the transmission through a public channel, as pointed out by one of the anonymous reviewers; therefore, the robustness of a secure image encryption scheme is also an essential performance merit. For testing the robustness of the proposed image encryption scheme, the noise-adding attack and the partial occlusion attack in the ciphertext domain are investigated.

(a) Noise-adding Attack: the ciphered and the de-ciphered images that are presented in Figure 11 are obtained based on the enhanced CIES-UBPRPD, in which the testing ciphertext images have been contaminated by adding with different degrees of salt-and-pepper noises. For performance analysis, we employ the widely used Peak Signal-to-Noise Ratio (PSNR) to evaluate the algorithm’s restoring ability. That is
$$\mathrm{PSNR}=10\times \log_{10}\frac{255^{2}}{MSE}(\mathrm{dB}),$$
where,
$$\mathrm{MSE}=\frac{1}{3\times M\times N}\sum_{i=0}^{M-1}\sum_{j=0}^{N-1}\sum_{k=0}^{2}{\left(O\left(i,j,k\right)-D\left(i,j,k\right)\right)}^{2},$$

*O* is the decrypted image obtained from the clean cipher image and *D* is the decrypted image obtained from the polluted cipher image. Generally, a higher PSNR indicates a better quality or ability of reconstruction. Table 4 shows the PSNRs of the proposed approach against the noise-adding attack.

(b) Partial-occlusion Attack. The ciphered and the de-ciphered images that are presented in Figure 12 are obtained based on the enhanced CIES-UBPRPD, in which the testing ciphertext images have been contaminated by occluding some parts of them (i.e., zeroing out those pixel values) that are depicted by the black segments. Similar to the noise-adding attack, PSNR is used to evaluate the proposed approach’s restoring ability against this attack, as shown in Table 5.

From Figure 11 and Figure 12, we can still recognize the de-ciphered images, even if the clean cipher images are contaminated by noise-adding or partial-occlusion attacks. This implies that the robustness of the enhanced CIES-UBPRPD is acceptable to practical image security applications.

## 6. Discussions and Conclusions

### 6.1. Discussions

In the past few years, combining both compression and encryption in a single algorithm to reduce the complexity is a new tempting approach for securing data during transmission and storage [14,15,16,17,18]. This new approach aims to extend the functionality of compression algorithms to achieve both compression and encryption simultaneously in a single process without an additional encryption stage. Employing the new combined simultaneous compression-encryption approach highly reduces the required resources for encryption (computational and power resources), according to [15] and [18]. Owing to such an attractive property, lots of works are devoted to this topic [19,20,21,22,23], some of them are also chaotic map based. In [22], we proposed three techniques for enhancing various chaos-based joint compression and encryption (JCAE) schemes. They respectively improved the execution time, compression ratio, and estimation accuracy of three different chaos-based JCAE schemes. However, all of the above-mentioned works are plain image independent. Therefore, for enhancing the security level further, how to design an effective plain Image dependent JCAE scheme is one of our future research directions.

Since steganography can also be utilized to conceal the private information, such as to provide privacy protection of medical images, as pointed out by one of the anonymous reviewers, besides Image Encryption and Decryption, Image Steganography is another worthy of noticing area in the field of Image Security. Interested readers are referred to the following informative writeups, although it is not within the scope of this work [24,25].

### 6.2. Conclusions

In this paper, we make detailed cryptanalysis on a published chaotic map-based image encryption system, where the encryption process is plaintext Image dependent. We show that some designing weaknesses make the published cryptosystem vulnerable to chosen-plaintext attack, and we then proposed an enhanced algorithm to overcome those weaknesses.

In summary, we use the SHA-256 hash value instead of the sum of each channel as a “plaintext feature”, so the enhanced CIES-UBPRPD has higher plaintext sensitivity than its original counterpart. Moreover, the newly proposed encryption process includes the cross-channel interaction, which can resist the Chosen Plaintext Attack that we launched. Since the SHA-256 hash value also serves as an external key, making the improved Keyspace reaches to 2^311^, which is larger than the effective Keyspace 2^120^ of the original CIES-UBPRPD. Besides the security and the robustness, we also take the execution time into account to provide a full performance baseline for comparing the original and the proposed image encryption algorithms, as suggested by one of the anonymous reviewers. Since the calculation amount of the enhanced CIES-UBPRPD is larger than that of the original CIES-UBPRPD, as shown in Table 6, the execution time of the proposed algorithm will be slightly slower than that of its original counterpart. Our implementation is conducted on Intel Core i7-8700 CPU @ 3.2GHz and 32GB RAM with Windows 10 OS, and written in Python.

The value of the proposed algorithm for practical usage in image security is justified, according to the security, the robustness, and the timing performance analyses. Finally, since the currently proposed chosen-plaintext attack can be only effective to all RGB-Colored images of size 256×256 pixels, finding an attack that can be applied to more general cases is one of the possible future extensions. We humbly hope that this paper will remind researchers to pay more attention when building their chaotic-based image encryption algorithms in the future.

## Figures and Tables

**Figure 1 entropy-22-00589-f001:**
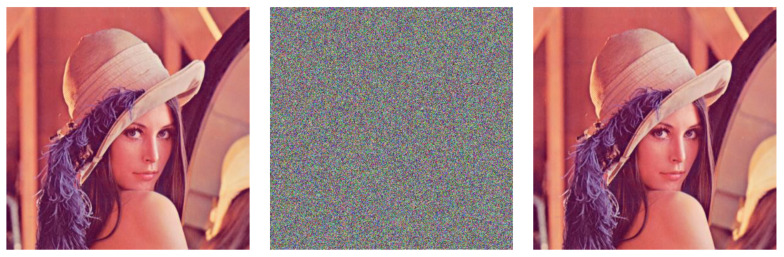
An Example of the Problem of Equivalent Classes in Keyspace.

**Figure 2 entropy-22-00589-f002:**
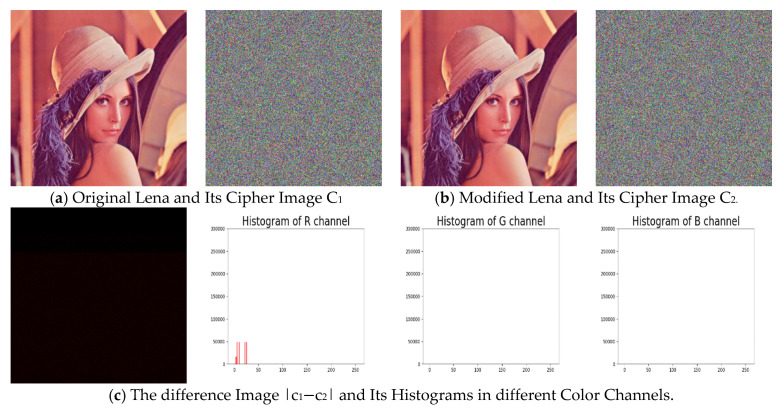
An Illustration Example of the Low Sensitivity to the Change of Plaintext Image.

**Figure 3 entropy-22-00589-f003:**
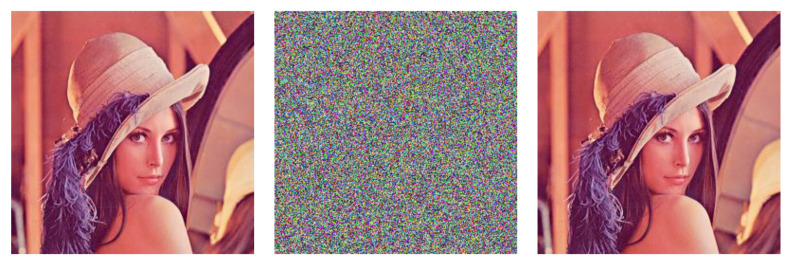
(Left) The Original Plaintext Image; (Middle) the Encrypted (or Ciphertext) Image; (Right) the Recovered Image After Launching the Proposed Chosen-plaintext Attack to the Ciphertext Image.

**Figure 4 entropy-22-00589-f004:**
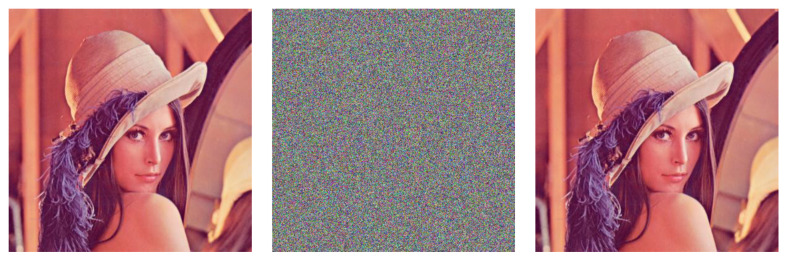
The Testing Results of the Modified Encryption Scheme: (left) the Plaintext Images; (middle) the Ciphertext Images; and, (right) the Recovered Images.

**Figure 5 entropy-22-00589-f005:**
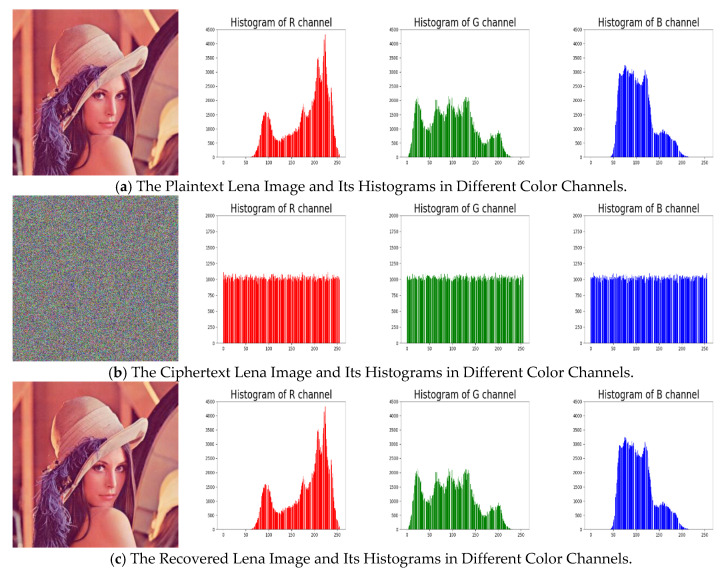
The Original/Encrypted/Recovered Lena Images and the Corresponding Histograms in Different Color Channels.

**Figure 6 entropy-22-00589-f006:**
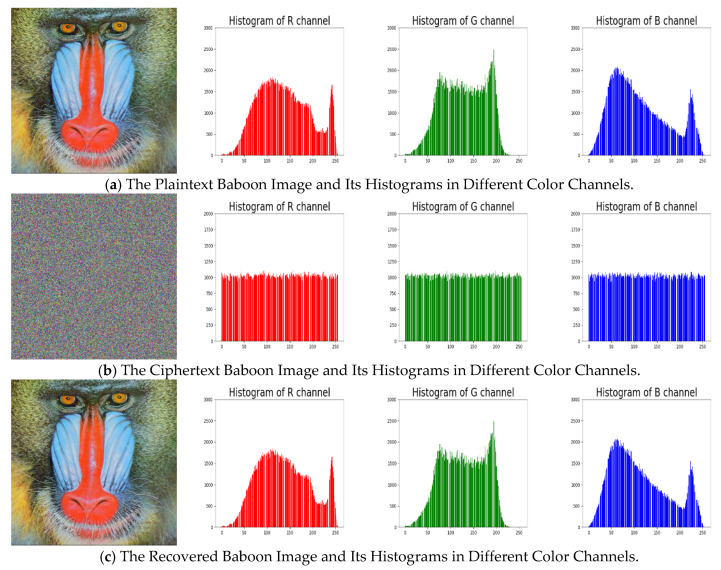
The Original/Encrypted/Recovered Baboon Images and the Corresponding Histograms in Different Color Channels.

**Figure 7 entropy-22-00589-f007:**
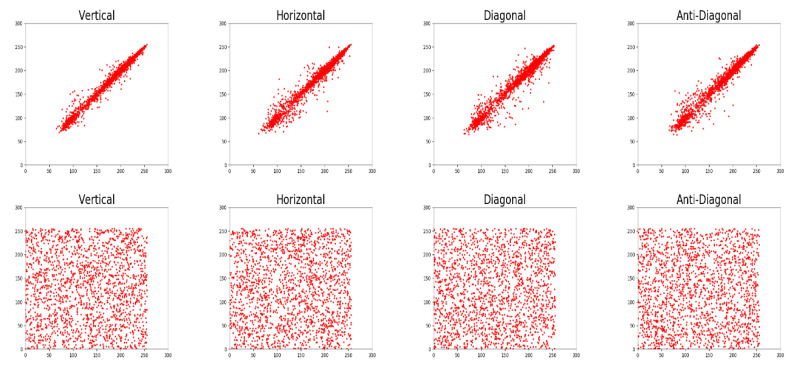
The Scatter Diagrams of Randomly Select 2000 Pairs of Adjacent Pixels at Four Specific Directions from the R Channel of (top) the Plaintext and (bottom) the Ciphertext Lena images.

**Figure 8 entropy-22-00589-f008:**
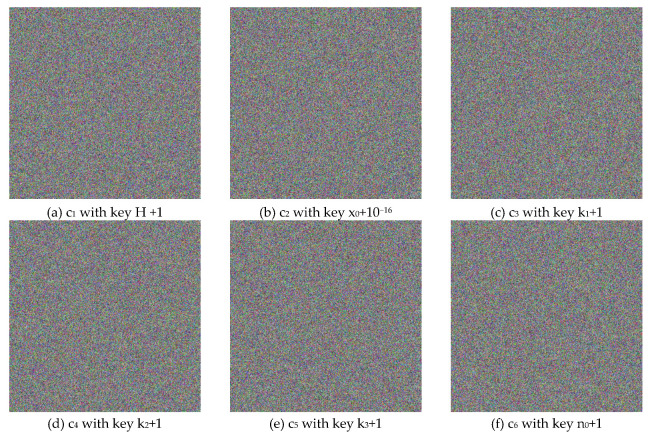
The Six Ciphertext Images Obtained by Changing Single Bit of Different Secret Key Parameters.

**Figure 9 entropy-22-00589-f009:**
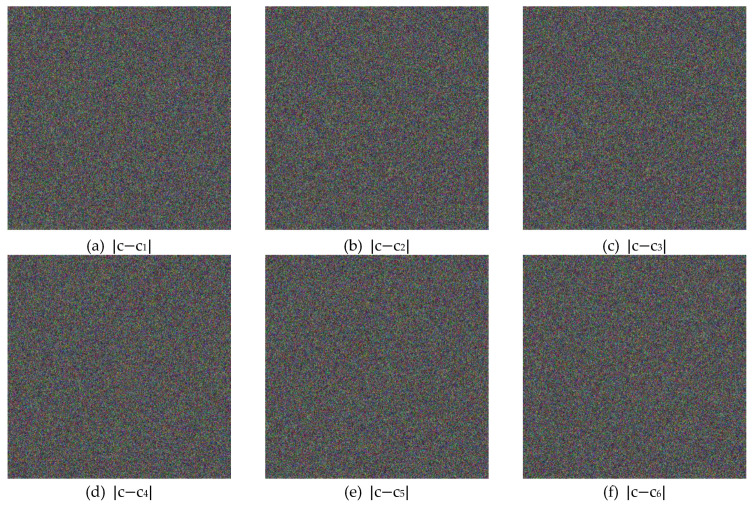
The Differential Results Between the Six Cipher Images given in Figure 8 and the Cipher Image has given in Figure 4b. Notice that, without notation confusion, the Original Ciphertext Lena Image is denoted as Image c, here.

**Figure 10 entropy-22-00589-f010:**
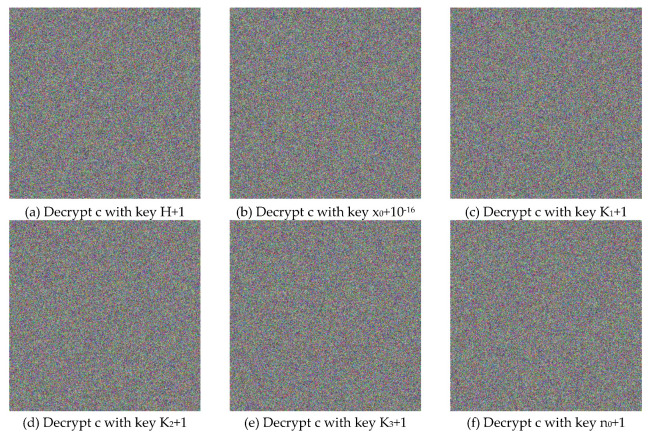
The Decrypted Images of Figure 4b with the 1-bit Difference Keys, mentioned in Sub-Section 5.2.4.

**Figure 11 entropy-22-00589-f011:**
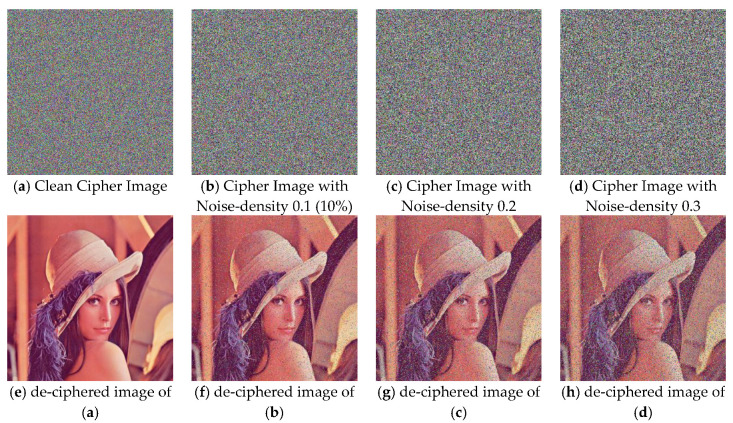
Robustness of the Proposed Approach Against the Noise-adding Attack.

**Figure 12 entropy-22-00589-f012:**
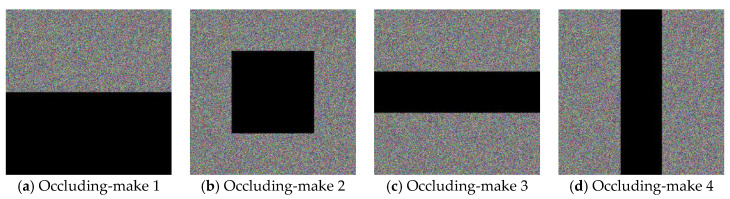
Robustness of the Proposed Approach Against the Partial-occlusion Attack.

**Table 1 entropy-22-00589-t001:** Number of pixels change rate (NPCR) and unified average changing intensity (UACI) Values Between the Two Cipher Images Given in Figure 2.

	R	G	B
NPCR (%)	86.5371	0	0
UACI (%)	4.8464	0	0

**Table 2 entropy-22-00589-t002:** Correlation Coefficients of the Plain/Cipher Lena Images.

		Plain Image			Cipher Image		
		R	G	B	R	G	B
Lena	V	0.9893	0.9823	0.9574	0.0015	−0.0017	−0.0023
	H	0.9797	0.9689	0.9325	−0.008	−0.0014	−0.0013
	D	0.9696	0.9554	0.9180	−0.003	−0.0011	−0.0011
	A	0.9777	0.9652	0.9252	−0.006	−0.0005	−0.0002
baboon	V	0.8659	0.7650	0.8808	0.0004	−0.0017	0.006
	H	0.9230	0.8654	0.9073	0.0005	0.0027	0.0019
	D	0.8543	0.7347	0.8398	0.0004	−0.0026	0.0014
	A	0.8518	0.7249	0.8424	−0.0015	−0.0017	0.002

**Table 3 entropy-22-00589-t003:** The Plaintext Sensitivity Analyzing Results, Measured in Terms of Averaged NPCR and UACI Values.

	NPCR (%)			UACI (%)		
	R	G	B	R	G	B
Lena	99.6094	99.6084	99.6096	33.4673	33.4630	33.4662
baboon	99.6075	99.6081	99.6086	33.4606	33.4646	33.4684
fruits	99.6081	99.6103	99.6095	33.4612	33.4620	33.4689
airplane	99.6094	99.6071	99.6101	33.4669	33.4595	33.4564
peppers	99.6099	99.6109	99.6092	33.4649	33.4641	33.4702

**Table 4 entropy-22-00589-t004:** Robustness of the Proposed Approach Against the Noise-adding Attack, in terms of Peak Signal-to-Noise Ratio (PSNR).

Added Noise Density	PSNR of the De-Ciphered Image
0.1 (10% of the image frame)	17.6748 (dB)
0.2	14.9988 (dB)
0.3	13.5375 (dB)

**Table 5 entropy-22-00589-t005:** Robustness of the Proposed Approach Against the Partial-occlusion Attack, in terms of PSNR.

Data Occlusion Loss	PSNR of the De-Ciphered Image
Pixel values in the bottom half = 0	11.6397 (dB)
Pixel values in the center square = 0	14.6680 (dB)
Pixel values in center row-rectangle = 0	14.6726 (dB)
Pixel values in center column-rectangle = 0	14.6285 (dB)

**Table 6 entropy-22-00589-t006:** The Timing Performance Comparison of the Original and the Proposed Approaches.

	Original CIES-UBPRPD	Enhanced CIES-UBPRPD
512 × 512 (image size)	3.3557 (s)	3.7944 (s)
256 × 256 (image size)	0.8355 (s)	0.9165 (s)

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
