# Peer review of "Cryptanalysis and Improvement of a Chaotic Map-Based Image Encryption System Using Both Plaintext Related Permutation and Diffusion"

_entropy, 2020, doi:10.3390/e22050589_

Round 1
Reviewer 1 Report
The authors in this paper analyze a published chaotic map-based image encryption system and propose an enhanced algorithm to overcome the vulnerability of the published cryptosystem. However, the authors should consider the following suggestions to improve the paper.
- We hold the view that the authors should explain the reasons for changing all array indexes starting from zero.
- It is recommended that the authors should offer a standard baseline,e. the performance of the published cryptosystem, so as to compare it with the enhanced algorithm proposed.
- Whether the image encryption system proposed by the authors is robust?Since a small change in the encrypted image may induce a strong distortion in the decrypted image and increase the difficulty of original image restoration, we suggest the robustness of the encryption system should be verified.
- In section 3.2, the authors present a chosen-plaintext attack that works if the size of all images equals to 256×256 pixels, however, they apply the proposed algorithm to testing images with the size of 512× There is a question about the process of image transformation.
- Some works about image security should be considered and added, such as 10.1016/j.compeleceng.2017.08.020, 10.1007/s11042-017-4946-9
Author Response
Comments and Suggestions for Authors Reviewer 1:
Reviewer 1’s comment 1: We hold the view that the authors should explain the reasons for changing all array indexes starting from zero.
Our reply:
Since the residue of integers modulo N belongs to the set {0,1,2, ..., N-1}, it’s a convention for signal processing and cryptographical society people start their indexes of an array from 0. Even though the representations look different from each other; actually, with proper index shifting (just as the case in this work and target reference), the effects of indexes starting from 0 is the same as that of starting from 1. Of course, if the software simulation is conducted on MATLAB, arrays with indexes started from 1 may be more convenient, but it is not a must-have condition.
Reviewer 1’s comment 2: It is recommended that the authors should offer a standard baseline,e. the performance of the published cryptosystem, so as to compare it with the enhanced algorithm proposed.
Our Reply:
We use the SHA-256 hash value instead of the sum of each channel as a "plaintext feature", so the enhanced CIES-UBPRPD has a higher plaintext sensitivity than its original counterpart. Moreover, the newly proposed encryption process includes the cross-channel interaction, which can resist the Chosen Plaintext Attack launched by us. Since the SHA-256 hash value also serves as an external key, making the improved Keyspace reaches to 2 311, which is larger than the effective Keyspace 2 120 of the original CIES-UBPRPD.
As suggested by anonymous reviewers, we also compared the execution time of the original and the enhanced CIES-UBPRPD. Since the calculation amount of the latter is larger than that of the former, as shown in the following table, the execution time of the enhanced CIES-UBPRPD will be slightly slower than that of the original CIES-UBPRPD. Our implementation is conducted on Intel Core i7-8700 CPU @ 3.2GHz and 32GB RAM with Windows 10 OS, and written in Python.
|
|
Original CIES-UBPRPD |
Enhanced CIES-UBPRPD |
|
512*512 (image size) |
3.3557 (sec) |
3.7944 (sec) |
|
256*256 (image size) |
0.8355 (sec) |
0.9165 (sec) |
In response to this insightful comment, a new paragraph is added in Section 6.2 for comparing the enhanced CIES-UBPRPD with its original counterpart.
Reviewer 1’s comment 3: Whether the image encryption system proposed by the authors is robust?Since a small change in the encrypted image may induce a strong distortion in the decrypted image and increase the difficulty of original image restoration, we suggest the robustness of the encryption system should be verified.
Our Reply:
As pointed out by one of the anonymous reviewers, cipher images can easily be polluted during the transmission through a public channel; therefore, the robustness of a secure image encryption scheme is also an essential performance merit. For testing the robustness of the proposed image encryption scheme, the noise-adding attack and the partial occlusion attack in the ciphertext domain are investigated, in the revision. The simulation results, including two figures and two tables, are presented in the newly added “Robustness Analyses” section of the revision.
We sincerely appreciate reviewer 1 for bringing this significant factor to our attention, which does assist us in justifying the value of the proposed approach in practical usage.
Reviewer 1’s comment 4: In section 3.2, the authors present a chosen-plaintext attack that works if the size of all images equals to 256×256 pixels, however, they apply the proposed algorithm to testing images with the size of 512× There is a question about the process of image transformation.
Our Reply:
In this work, we choose 512x512 as the testing Image size merely because most of the available benchmarking Images are of that size. On the other hand, the reason why we take 256x256 as the size of Images to be attacked comes from the fact that this choice will simplify the associated computations a lot. For example, in equation (6), if we set M=N=256, then only the last 8 Least significant bits of the Cat-map’s parameters matter. Under this consideration, the feasible solution can be found by brute-force searching.
As suggested by Bruce Schneier [B1], in academic cryptography, a weakness or a break in a scheme is usually defined quite conservatively: it might require impractical amounts of time, memory, or known plaintexts. It also might require the attacker to be able to do things many real-world attackers cannot. For example, the attacker may need to choose particular plaintexts to be encrypted or even to ask for plaintexts to be encrypted using several keys related to the secret key. In other words, to break the original CIES-UBPRPD, a specific choice of 256x256 as the attacking image size is acceptable. To clarify this confusion in the original manuscript, we add a new paragraph at the beginning of sub-section 3.2 of our revision.
[B1] Schneier, Bruce , "A Self-Study Course in Block-Cipher Cryptanalysis". Cryptologia. 24 (1): 18–34, Jan. 2000; doi:10.1080/0161-110091888754.
Reviewer 1’s comment 5: Some works about image security should be considered and added, such as 10.1016/j.compeleceng.2017.08.020, 10.1007/s11042-017-4946-9
Our Reply:
Thanks to Reviewer 1 for reminding us that Image Encryption is not the only exciting topic in the field of Image security. To enhance the discussion of the Image security-related researches, we add a sub-section (Section 6.1) in the revision. In which, the topic of joint Image Compression and Encryption is introduced. Also, as suggested by one of the anonymous reviewers, a short paragraph for referencing the following two informative works about Image Steganography have been included as well.
[A1] Xin Liao, Sujing Guo, Jiaojiao Yin, Huan Wang, Xiong Li, and Arun Kumar Sangaiah, “New cubic reference table based image steganography,” Multimedia Tools and Applications volume 77, pages 10033-10050, June 2017 DOI 10.1007/s11042-017-4946-9.
[A2] Xin Liao, Jiaojiao Yin, Sujing Guo, Xiong Li, and Arun Kumar Sangaiah, “ Medical JPEG image steganography based on preserving inter-block dependencies,” Computers & Electrical Engineering 67, August 2017; DOI: 10.1016/j.compeleceng.2017.08.020.
Reviewer 2 Report
The proposed cryptanalysis method is not clear and consider only one case not general case. The paper in [10] considered only one round of plaintext encryption, so is it enough to consider their work weak after 16 round for example. Also, the improvement presented by the authors did not compared with the original paper to clear shown the difference in results obtained. Some other things to mention here:
1- the introduction is too long with topics far from the main topic of this work.
2- figures and tables labels and caption need to be more specific such as figure 5 and figure 6 instead of figure 5.1 and figure 5.2. as in pages 14 and 15.
Author Response
Comments and Suggestions for Authors Reviewer 2:
Reviewer 2’s comment 1: The proposed cryptanalysis method is not clear and consider only one case not general case.
Our Reply:
As suggested by Bruce Schneier [B1], in academic cryptography, a weakness or a break in a scheme is usually defined quite conservatively: it might require impractical amounts of time, memory, or known plaintexts. It also might require the attacker to be able to do things many real-world attackers cannot. For example, the attacker may need to choose particular plaintexts to be encrypted or even to ask for plaintexts to be encrypted using several keys related to the secret key. Furthermore, it might only reveal a small amount of information, enough to prove the cryptosystem imperfect but too little to be useful to real-world attackers. Finally, an attack might only apply to a weakened version of cryptographic tools, like a reduced-round block cipher, as a step towards breaking the full system. In other words, to break a crypto-scheme considering one specific case is good enough.
Thanks to Reviewer 2 for reminding us to include this point into the revision. To emphasize this important principle for cryptanalysis, we add a new paragraph at the beginning of sub-section 3.2 of our revision.
[B1] Schneier, Bruce , "A Self-Study Course in Block-Cipher Cryptanalysis". Cryptologia. 24 (1): 18–34, Jan. 2000; doi:10.1080/0161-110091888754.
Reviewer 2’s comment 2: The paper in [10] considered only one round of plaintext encryption, so is it enough to consider their work weak after 16 round for example.
Our Reply:
We think there is some misunderstanding about the encryption process of the improved CIES-UBPRPD. Actually, both the permutation and diffusion stages are run only one round, just like that of the original CIES-UBPRPD. For example, the ciphertext Lena images shown in Fig. 1, Fig. 2(b), and Fig.3 (middle) are all obtained by running the encryption process one round only. Besides the keyspace shrinking, the major problem of the original CIES-UBPRPD lies from its low sensitivity to the change of plaintext design weakness, mentioned in sub-section 3.1 (b). And the effect of this weakness has been proven by launching the chosen-plaintext attack, presented in Section 3.2. To clarify that the encryption of the proposed CIES-UBPRPD is a one-round process, a short sentence for emphasizing this fact is added in the last paragraph of sub-section 2.2.
Reviewer 2’s comment 3: Also, the improvement presented by the authors did not compared with the original paper to clear shown the difference in results obtained.
Our Reply:
We use the SHA-256 hash value instead of the sum of each channel as a "plaintext feature", so the enhanced CIES-UBPRPD has a higher plaintext sensitivity than its original counterpart. Moreover, the newly proposed encryption process includes the cross-channel interaction, which can resist the Chosen Plaintext Attack launched by us. Since the SHA-256 hash value also serves as an external key, making the improved Keyspace reaches to 2 311, which is larger than the effective Keyspace 2 120 of the original CIES-UBPRPD.
As suggested by anonymous reviewers, we also compared the execution time of the original and the enhanced CIES-UBPRPD. Since the calculation amount of the latter is larger than that of the former, as shown in the following table, the execution time of the enhanced CIES-UBPRPD will be slightly slower than that of the original CIES-UBPRPD. Our implementation is conducted on Intel Core i7-8700 CPU @ 3.2GHz and 32GB RAM with Windows 10 OS, and written in Python.
|
|
Original CIES-UBPRPD |
Enhanced CIES-UBPRPD |
|
512*512 (image size) |
3.3557 (sec) |
3.7944 (sec) |
|
256*256 (image size) |
0.8355 (sec) |
0.9165 (sec) |
In response to this insightful comment, a new paragraph is added in Section 6.2 to compare the enhanced CIES-UBPRPD with its original counterpart.
Reviewer 2’s comment 4: Some other things to mention here:
1- the introduction is too long with topics far from the main topic of this work.
2- figures and tables labels and caption need to be more specific such as figure 5 and figure 6 instead of figure 5.1 and figure 5.2. as in pages 14 and 15.
Our Reply:
- As suggested, the introduction Section of the revision has been shortened and focused more on the main topic of our work.
- The captions of involved Figures and Tables of the revision have been modified as suggested. Moreover, the resolution of all Figures and Tables has been enhanced as well.
Round 2
Reviewer 1 Report
The authors have revised the paper according to the comments, and I think it is acceptable now.
Reviewer 2 Report
Thanks for your revised work. The addition parts is very important and comparison with previous work clearly described.